# Why are listeners sometimes (but not always) egocentric? Making inferences about using others' perspective in referential communication

J. Jessica Wang[1]*, Natalia Ciranova[1], Bethany Woods[1], Ian A. Apperly[2]*

**1** Department of Psychology, Lancaster University, Lancaster, United Kingdom, **2** School of Psychology, University of Birmingham, Edgbaston, United Kingdom

\* j.wang40@lancaster.ac.uk (JJW); i.a.apperly@bham.ac.uk (IAA)

**Data Availability Statement:** All relevant data are within the manuscript and its Supporting Information files.

## Abstract

Theory of Mind (ToM) is the ability to understand others' mental states, and that these mental states can differ from our own. Although healthy adults have little trouble passing conceptual tests of ToM (e.g., the false belief task [1]), they do not always succeed in using ToM [2,3]. In order to be successful in referential communication, listeners need to correctly infer the way in which a speaker's perspective constrains reference and inhibit their own perspective accordingly. However, listeners may require prompts to take these effortful inferential steps. The current study investigated the possibility of embedding prompts in the instructions for listeners to make inference about using a speaker's perspective. Experiment 1 showed that provision of a clear introductory example of the full chain of inferences resulted in large improvement in performance. Residual egocentric errors suggested that the improvement was not simply due to superior comprehension of the instructions. Experiment 2 further dissociated the effect by placing selective emphasis on making inference about inhibiting listeners' own perspective versus using the speaker's perspective. Results showed that only the latter had a significant effect on successful performance. The current findings clearly demonstrated that listeners do not readily make inferences about using speakers' perspectives, but can do so when prompted.

## Introduction

Theory of Mind (ToM) is the ability to understand others' mental states, and that these mental states can differ from our own. This ability is fundamental for navigating the social world in which we live. Without ToM, it would be impossible to achieve mutual understanding or even interpret others' simple actions, e.g., a friend's point to a peppermill on the dinner table. Although typically-developed adults have little trouble passing conceptual test of ToM, there is much variability in their propensity to *use* what they know about others' mental states in ongoing communication (e.g., [2,3]). Instances of faux pas are regularly seen in daily life, and are often due to the actor or speaker not fully accounting for others' mental states. For example,

**Funding:** Experiment 1 was supported by a research grant funded by the Economic & Social Research Council to IA (reference: ES/J012238/1). https://esrc.ukri.org Experiment 2 was partly supported by a research grant funded by the British Academy and the Leverhulme Trust to JW (reference: SG162831). https://www. thebritishacademy.ac.uk Neither of the funders had any role in study design, data collection and analysis, decision to publish, or preparation of the manuscript.

**Competing interests:** The authors have declared that no competing interests exist.

the result of the Great British Bake Off final in 2017 was revealed hours ahead of broadcast due to new judge Prue Leith forgetting that she was in a time zone hours ahead of the UK and congratulated the winner on Twitter prematurely. Real life egocentric errors that carry higher stakes can also be seen in the business and political domains (e.g., In the US presidential election campaign in 2016, Hillary Clinton calling half of Trump's supporters "a basket of deplorables", wiping out her chances of winning over their votes). Instances like these highlight that we cannot assume successful or consistent ToM-*use* among people who would clearly pass standard conceptual tests of ToM. Egocentric errors are frequently observed in real life, not only in face to face communication, but also over email communications [4]. On the other hand, such errors are far from ubiquitous: communicators frequently display sensitivity to perspective differences (e.g., [5–7]). This leads to a puzzle about why people only sometimes use their ToM abilities. In the present study we investigated the possibility that listeners have a tendency to overlook some of the inferential steps required to fully account for a speaker's perspective. We employed a task that normally produces consistently high rates of egocentric errors (e.g., [2,3]), therefore allowing space for improvement in ToM-use.

Referential communication tasks provide an ideal context to systematically capture communicators' successes and failures in using ToM (e.g., [2,3,5,6,8–12]). In order for communicators to correctly understand others' reference, they need to account for the common ground they shared with their communicative partner while avoid referring to information privileged to themselves [13,14]. The premise of these tasks critically tests the degree to which communicators are able to *use* what they know about others' mental states during language comprehension (e.g., [2,3,5,6] and language production (e.g., [11,15]). A widely-used task in this domain is the director task, which requires participants to take the role of listeners and follow instructions delivered by a director. Critically, there is a discrepancy between the participants' perspective and the director's perspective, as they each have a unique view of the same grid (see Fig 1). Some of the slots on the grid are open to both the director's side and the participants' side, therefore any objects placed in these slots are in their common ground. In contrast, some

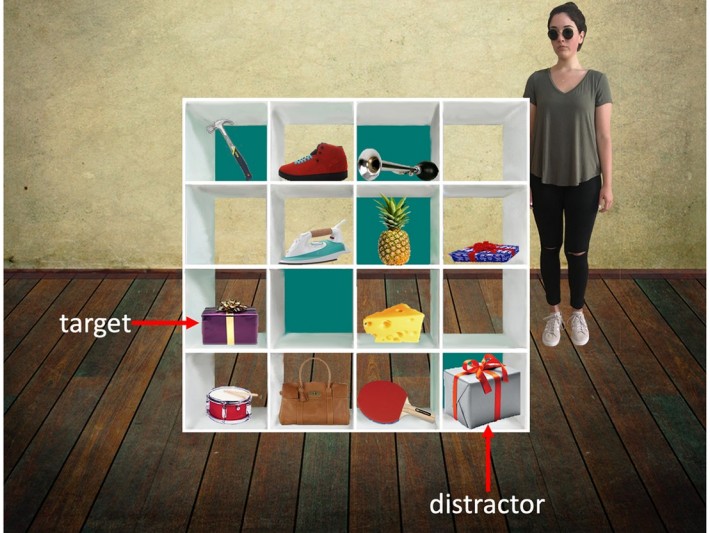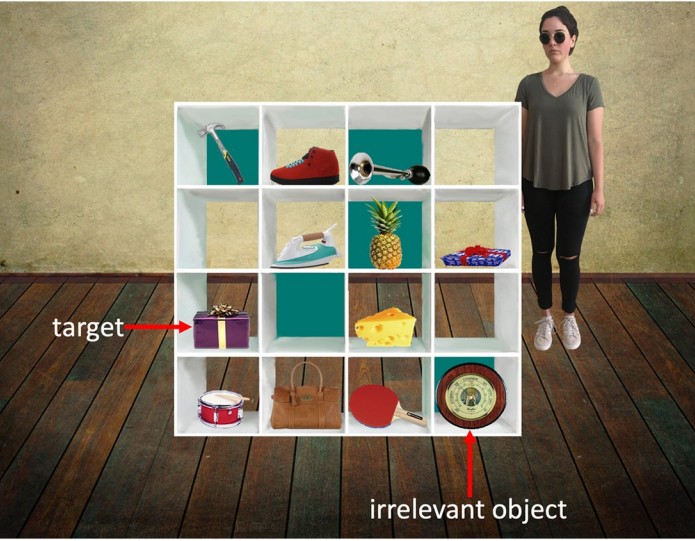

**Fig 1. Examples of the grid display.** An example of the experimental condition is shown on the left, with the control condition on the right. A critical instruction to accompany this display would be "nudge the large present one slot up". The only difference between the experimental condition and the control condition is that the experimental condition contains a "distractor" (in this case, the largest present from participants' view), which competes with a "target" (in this case, the largest present from director's view) to be the best-fitting referent for the director's critical instructions. In the control condition, the distractor is replaced by an "irrelevant object" (in this case, the barometer from participants' view), which does not compete with target to be the best fitting referent.

of the slots on the grid are blocked from the director's view, therefore any objects placed in these slots are in the participants' privileged ground, as these objects cannot be seen by the director. In order to correctly interpret the director's utterances, participants need to realise that the director's perspective content is different to their own. In particular, participants need to realise that the director has a restricted view of the grid, and that she can only see the objects in the common ground. This requirement is heavily emphasised in the introductory procedures to the task (see below for more detail). Critically, calculating the director's perspective is not sufficient to succeed on the task: participants must also interpret the director's utterances according to her perspective, not the participants' own perspective. To do so participants must set aside or inhibit their own perspective, and realise that the director's perspective means she cannot be referring to some objects. Failure to ignore their own perspective and successfully apply the director's perspective leads participants to commit egocentric errors in referent selection and/or show egocentric tendencies in response time and eye movements. Given the importance of ToM-use in social cognition, variations of the director task have been widely employed to study the developmental changes [12,16–18], cross-cultural differences and similarities [19–21], individual differences in social and cognitive functioning [22,23], and neural-underpinnings of ToM-use [16,24,25].

A striking feature of the director task is the high rates of egocentric errors frequently observed [2,3,12,16,18]. What is more, attempts to reduce rates of egocentric errors have often been strikingly unsuccessful. Keysar et al. [3] attempted to maximise the saliency of a director's perspective by implementing a number of measures. For instance, they had the participating listeners hide a referential competitor in an opaque bag themselves to highlight the director's ignorance; they had the participants swap roles with the director during practice to highlight her perspective; they even assigned the director a false belief so that the director held a distinctively different belief to the participants. However, none of these measures made the listeners more successful at using the director's perspective. Similar degrees of egocentrism were observed when the director had a salient false belief about the content of listeners' privileged ground versus when she was simply ignorant about it. This suggests that saliency of the director's perspective content is unlikely to account for the high rates of egocentric errors observed. Legg et al. [9] further confirmed that the level of difficulty in inferring a director's perspective content did not have a significant effect on the degrees of egocentrism listeners displayed. In this study, listeners were no more successful in avoiding egocentric errors when the inferential process merely concerned computing her visual access (level-1 visual perspective) versus the way in which objects appeared to her (level-2 visual perspective). These findings suggest that the inference required to calculate others' perspective content does not have a clear role in accounting for failure in ToM-use.

A mindreading model put forward by Apperly [26] suggests that mindreading involves three constituent stages: *calculating* others' mental states, *storing* the calculated mental state, and *using* the calculated mental state to explain, predict, or interpret behaviour. Keysar et al [3] and Legg et al [9] demonstrated that unsuccessful mindreading is unlikely to arise from the calculation stage. Zhao et al. [18] studied the storage stage and showed that 8- and 10-year-old children committed more egocentric errors when they were required to remember whether an object belonged to the director's perspective or their own privileged perspective. This suggests that egocentrism could be at least partly attributed to failure in the storage stage. However, in the director tasks where high rates of egocentric errors were observed in adults [2,3], there was no explicit requirement to hold the director's perspective in mind, because unlike the Zhao et al. [18] study, the director's perspective could always be inferred from the visual information available at the time of need. Therefore some of the egocentric errors previously observed are likely to arise from the final *use* stage. Specifically, having calculated the director's perspective, participants need to do two things to use it successfully. Firstly, participants must ensure that

they are guided by the director's perspective rather than their own. Secondly, participants need to work out precisely how the director's perspective constrains reference. It is worth noting that in standard versions of the director task, the director's perspective and the participants' perspective differ in their visual access to various objects, which corresponds to level-1 visual perspective judgements of *what* can be seen by others. The calculation of level-1 visual perspective content has been shown to be relatively effortless (e.g., [27,28]) therefore participants are unlikely to have difficulty in calculating the content of the director's perspective. Instead, the critical process involved in this inferential step likely lies in inferring the implications of having different visual perspectives to the director. In the context of the director task, the implication is that the director's instructions can be interpreted differently from participants' own perspective versus the director's perspective (e.g., the 'large present' in Fig 1 could be interpreted as corresponding to different objects from the director's perspective versus the participants' perspective). It is possible that the high rates of egocentric errors previously observed (e.g., [2,3]) resulted from failure to achieve either or both of these steps. Such perspective-taking failures are also seen in everyday language production. In the example we described at the start of this paper, Bake-off's Prue Leith's premature congratulation to the winner of the show is unlikely caused by a difficulty in understanding the concept of time zones, or an ability to understand that she has knowledge about the outcome of the show that is privileged to her but not the show's viewers. However, when it mattered, she still failed to account for her viewers' perspectives, spoiling the show's finale. Instances of perspective-taking failures highlight the difficulty people encounter in *using* what we know about others' mental states. Understanding these processes would cast new light on the nature of how people *use* mindreading information [26]. It is clear that having the conceptual understanding that people can have different perspectives does not guarantee successful *use* of such information. Therefore it is critical to examine the potential sub-processes involved in ToM-use.

We employed a variation of the computer-based director task that has been shown to produce high rates of egocentric errors [2]. In order to encourage participants to take the inferential steps required for successful ToM-use, we systematically manipulated the overt instructions to emphasise the need for participants to inhibit their own perspectives and to infer the way in which a director's perspective constrains reference. In Experiment 1, prompts for the two steps associated with self-perspective inhibition and other-perspective-use were presented together. If the high rates of egocentric errors previously observed are driven by participants' oversight of the inferential steps required, then the provision of an introductory example of the full chain of inferences should significantly reduce the rates of egocentric errors observed. Experiment 2 further investigated whether both prompts are required, and whether they need to be given sequentially.

## Experiment 1

The current experiment compared ToM-use performance following two version of overt instructions: 1. a director's perspective was made clear, and participants were instructed to take her perspective into account during the task. 2. exactly the same instruction as version 1, with the addition of an example of the full chain of inference required to successfully use the director's perspective. The example instructs participants not to use their own perspective, and to use the director's perspective to interpret her utterances.

### Method

**Participants.**   Sixty-eight participants (10 males, mean age 19.16 years, age range 18 to 23 years) gave informed consent to participate in the study and were tested by the same

experimenter at the University of Birmingham. Ethical approval has been granted by the Ethics Committee at the University of Birmingham (reference: ERN_09–719). The individual whose image featured as the director in this manuscript has given written informed consent (as outlined in PLOS consent form) to publish their image. The sample size required to detect an interaction between condition and task instruction was calculated using G*Power. Fifty-two participants were required with power set to 0.8, effect size f set to 0.2. As this is a novel effect, we tested 68 participants to ensure that we are able to detect the effect. Participants were given course credit or a small honorarium as reward (the form of reward was not recorded for this experiment, hence could not be included as a random effect in the analysis). Three participants were replaced as two of the participants self-reported strategies unrelated to ToM-use, and the third participant attributed conflict between self and other perspectives to computer fault.

**Design & procedure.**    We manipulated the overt instruction so that the way in which a director's perspective constrains reference was either made explicit to the participants via the overt instructions or not. A 2 x 4 x 2 mixed design was employed with condition (control, experimental) and magnitude of common ground (3, 5, 7, 9) as within-participant variables, and task instruction (with-example, without-example) as a between-participant variable. The only difference between the experimental condition and the control condition is that the experimental condition contains a "distractor", which competes with a "target" to be the best-fitting referent for the director's critical instructions (see Fig 1 for an example). In the control condition, the distractor is replaced by an "irrelevant object", which does not compete with target to be the best fitting referent. It is crucial to include such a control condition, as the processing cost associated with the control condition provides a baseline measure of the processing demands associated with visual search, speech processing, instructions following without perspective-taking, and object selection. Since the experimental condition and the control condition only differed by one object: the distractor versus irrelevant object, we can infer that any additional processing cost observed in the experimental condition compared to the control condition would reflect demand of perspective-taking, as opposed to processing visual stimuli, speech, instructions following, or object selection. The magnitude of common ground, which referred to the number of open slots on a grid, was manipulated in another series of unpublished studies. The with-example and without-example task instructions were both delivered as a combination of spoken instructions, images, and experimenters' actions. The only difference between the two sets of instructions was that the with-example condition included an example to illustrate the way in which the director's perspective should be used to interpret her utterances. The exact wording for the task instructions and their accompanying images can be found in Table 1.

The task instruction was followed by 2 practice images and 32 test images, presented in 4 test blocks. When an image appeared, participants had 5000ms to examine the image before hearing 3 to 5 instructions from the director, one of which was a critical instruction. A total of 128 instructions were presented, with 32 critical instructions. The critical instructions were "nudge the [scalar adjective] [noun] one slot [directional word]" (for the complete list of critical instructions, see Appendix A). Relational expressions were employed to maximise the likelihood of observing egocentric errors and the potential increase in successful ToM-use. The remaining 96 instructions were fillers, 32 of which contained scalar adjectives (14 of the scalar adjectives were redundant adjectives, included to minimise the likelihood for scalar adjectives to signal the need to take the director's perspective), and 16 contained non-scalar adjectives (e.g., blue, included to minimise the likelihood for adjectives to signal the need to take the director's perspective), and the remaining 48 filler instructions were simple noun phrases. All sentences were spliced together from individually recorded words to eliminate the possibility for participants to use co-articulation to identify a referent prior to the onset of the adjective or noun. If participants did not respond within 4000ms from the onset of the adjective (or noun

**Table 1. Instruction wording in Experiments 1 and 2.** All instructions in quotation marks were spoken, contents in parenthesis were acted out by an experimenter. The only difference between various conditions was the example given on Slide 3.

| Slide | Spoken instruction | Accompanying image |
|---|---|---|
| 1 | **E1 & E2:** "In this experiment, you will see a director, like the one shown on the screen. You will also see a 4x4 shelf posited between you and her. There will be some objects on the shelf. The director will give you instructions to move some of the objects around. <br><br> Some of the slots on the shelf are blocked from the director's point of view, and she does not know about the objects in those slots. Therefore she cannot ask you about those object. You would have to take this information into account when nudging the objects" | |
| 2 | **E1 & E2:** "In the next 20–25 minutes, you will see a lot of pictures like this. This is the shelf I was talking about, there are 16 possible locations for objects to go. You will notice that some of the slots have green backgrounds. You are able to see the objects in these slots. However, the director is standing behind the shelf, on the other side. Therefore she doesn't see and doesn't know about any objects placed in those slots (point to the five blocked slots). Since she doesn't know about these objects, she cannot possibly ask you to move any of them." |  |
| 3 | **E1 with-example condition & E2 other-explicit-self-explicit condition:** "For example, if she asks you to "nudge the short torch one slot left", although you might be tempted to move this object (point to the shortest torch), this object isn't actually available to her. Therefore she cannot be talking about this object, she must be talking about this object (point to middle torch) instead, because this is an object that she can see and can talk about. Does this make sense?" <br><br> **E2 other-explicit- self-not-explicit condition:** "For example, if she asks you to "nudge the short torch one slot left", she must be talking about this object (point to middle torch). Because this is an object that she can see and can talk about. Does this make sense?" <br><br> **E2 other-not-explicit-self-explicit condition:** "For example, if she asks you to "nudge the short torch one slot left", although you might be tempted to move this object (point to the shortest torch), this object isn't actually available to her. Therefore she cannot be talking about this object. Does this make sense?" <br><br> **E1 without-example condition & E2 other-not-explicit- self-not-explicit condition:** "For example, if she asks you to "nudge the short torch one slot left", you would have to consider her perspective when you follow her instructions. Does this make sense? |  |
| 4 | **E1 & E2:** "To make sure everything is clear to you, this is how the shelf looks from where the director is standing. Please keep her perspective in mind when you follow her instructions." |  |

where the instruction did not contain adjectives), then the trial timed out, and the next instruction was played or the next grid-image shown. As in Apperly et al. [2], participants responded with a computer mouse, by performing a "drag and drop" motion as if moving the selected object from one slot to another. Object selection accuracy and response time were based on the first mouse click participants performed following an instruction. Participants were informed that we were interested in their first mouse click, therefore they should consider carefully before making a mouse click. Interest areas were drawn around each slot on the grid, therefore a mouse click within the slot in which a correct object was positioned would qualify as a correct response. Response times were calculated from the onset of an adjective or noun until the first mouse click response.

Half of the images corresponded to the experimental condition, the other half corresponded to the control condition. In the experimental condition, the item that best fitted the director's description on a critical instruction differed from the director's point of view versus the participants' point of view. For example, when the director asked for the "large present", the item she referred to was the purple present in the left panel of Fig 1 as it is the larger of the two presents available to her ("target" hereafter). However, the item that best fitted the director's description from the participants' point of view was the white present ("distractor" hereafter). In order to correctly select a target, it was essential that participants utilized perspectival information to resolve reference. The control condition was identical to the experimental condition apart from that the distractor was replaced by an irrelevant item which did not compete as a potential referent from the participants' perspective (e.g., a barometer, see right panel of Fig 1). Participants saw the grid images associated with both the experimental condition and its counterpart in the control condition. The grid images were positioned at least a full block of 8 trials apart, and in different halves of the experiment, minimising the possibility for the critical difference between the conditions to be recognised by participants. No participant was able to describe the critical difference between the two conditions during debrief.

## Results

Trial level data and the R code used in the analyses from the current study can be found in the Supplementary Materials. We calculated the *percentage of egocentric errors* for each participant in each condition. An egocentric error refers to a response error of selecting the distractor rather than the target in the experimental condition. Selection of the irrelevant object in the control condition provides a baseline for erroneous selections in the absence of direct competition between the participants' and the director's perspectives on a closely matched grid image. Across the two experiments, only 2 of such selection errors were observed in the control condition. Selections of objects or spaces that are not distractors or targets were excluded from all analyses, as these errors are rare, and it is difficult to interpret the cause of such errors. Trials with response timeout were excluded prior to analysis, leading to exclusion of 4.41% of the critical trials from the current experiment. The considerably lower error rate in the control condition compared to the experimental condition lead to unequal variance between the two conditions, which made it questionable to include condition as a factor in an omnibus analysis. Therefore our analyses focus on percentage egocentric errors on the experimental conditions (for descriptive statistics, see Table 2). The overall rate of egocentric errors could reflect incorrect responses prior to participants' first correct response, or participants' consistency in using the director's perspective, or to some combination of these factors. These factors are separately informative about how instructions affect performance, therefore we examined the effect of instructions on overall egocentric error rate, number of trials to first correct response, and error rate following first correct response.

**Table 2. Descriptive statistics for Experiments 1 and 2.**

| E1 | | with-example | | without-example | |
|---|---|---|---|---|---|
| **Egocentric error (%)** | | **5.48** | | **55.84** | |
| SD | | 14.18 | | 34.42 | |
| **Number of trials to first correct response** | | **1.44** | | **4.58** | |
| SD | | 0.73 | | 3.34 | |
| **Error rate following first correct response (%)** | | **9.03** | | **50.58** | |
| SD | | 15.28 | | 33.74 | |
| E2 | | other-explicit | | other-not-explicit | |
| | | self-explicit | self-not-explicit | self-explicit | self-not-explicit |
| **Egocentric error (%)** | | **9.05** | **13.03** | **21.04** | **37.67** |
| SD | | 17.35 | 26.02 | 31.06 | 40.64 |
| **Number of trials to first correct response** | | **1.48** | **2.32** | **3.16** | **4.12** |
| SD | | 1.00 | 2.54 | 3.00 | 3.60 |
| **Error rate following first correct response (%)** | | **15.28** | **14.96** | **28.97** | **35.71** |
| SD | | 17.24 | 24.56 | 31.59 | 35.66 |

The combined results from the number of trials to first correct response and the error rates following first correct response can also help address an alternative account for any positive effect that the with-example condition might have on performance. Recall from the introduction that previous studies of perspective-taking during referential communication show a puzzling mixture of very good versus relatively poor performance. A potentially simple explanation of this pattern would be that studies demonstrating good performance simply employed clearer task instructions enabling more participants to understand that they should use their ToM abilities. This explanation would tell us something about the importance of instructions, but nothing about the underlying processes of ToM-use. If such an account were correct, then the with-example condition should not only require fewer trials until participants reach their first correct responses, but also show floor-level error rates following the first correct response, because superior instructions had removed the principal source of errors. To verify this alternative account, we will additionally compare the error rates following first correct response against zero. Response times from trials with correct responses were reported in the Supplementary Materials. Response time data should be interpreted with caution as these reports contain relatively small number of trials, due to exclusion of a large number of erroneous trials.

A generalized linear mixed effects model was fitted to egocentric errors using the glmer() function from the lme4 package in R [29]. The fixed effects were magnitude of common ground (3 slots, 5 slots, 7 slots, 9 slots), and task instruction (with, without explicit instruction to inhibit self-perspective and use the director's perspective). Both fixed effects were included as both main effects and interactions in all models. Both fixed effects were coded with contrast coding, specifically deviation coding, where each level is compared to a grand mean. Participant and grid image were included as random effects. Our models for Experiment 2 additionally included experimenter and reward as random effects. This was not possible here, as all participants were tested by one experimenter, therefore experimenter was not entered as a random effect. Information on the form of reward participants received was not recorded at the time, therefore reward was not entered as a random effect. We attempted to fit models with maximal random effect structure to all models [30]. The maximal model included intercepts from both random effects, and random slopes for magnitude of common ground by participant, task instruction by grid image. The fitted model did not contain random slots for the

**Table 3. Summary of mixed models from Experiment 1.**

|  | β | SE | χ2 | df | p |
|---|---|---|---|---|---|
| **Egocentric error** | | | | | |
| instruction | -5.12 | 0.83 | 34.54 | 1 | < .001 |
| mag | -0.53 | 0.43 | 1.48 | 1 | 0.224 |
| instruction*mag | -0.73 | 1.08 | 0.45 | 1 | 0.505 |
|  |  |  | t | df | p |
| **Number of trials to first correct response** | | | | | |
| instruction |  |  | 5.12 | 32.5 | < .001 |
|  |  |  | t | df | p |
| **Error rate following first correct response** | | | | | |
| instruction |  |  | -6.41 | 42.1 | < .001 |

magnitude of common ground by participant. The fitted model was used to determine the statistical significance of a given main effect or interaction by removing one main effect or interaction term from the fitted model at a time, and comparing the models with versus without a given effect. This comparison was conducted through the anova() function, which is suitable for comparing the variance for one or more fitted model objects.

The number of trials to first correct response and error rate following first correct response were aggregated by grid image and participant, therefore it was not possible to include these terms in mixed models. As there was no viable random effect to be included in the models, independent t-tests were carried out for these two dependent variables (for a summary of all analyses for the current experiment, see Table 3).

An effect of instruction was found in percentage egocentric error, number of trials to first correct response, and error rate following first correct response (see Fig 2). Participants' performance was significantly better across these measures when the way in which the director's perspective constrains reference was made explicit through a simple example in the instruction. The linear mixed effects models analysis on response times revealed a significant effect of condition (experiment > control), $\chi^2 = 9.85$, df = 1, $p = .002$, along with a significant interaction effect between condition and magnitude, $p = .017$. The full analysis can be found in the Supplementary Materials. To verify the possibility that the with-example condition merely clarified the instructions rather than improved ToM-use, the error rates following first correct response were tested against a floor-level. Error rates following first correct response in both conditions were significantly higher than floor-level ($ts > 3.54$, $ps < .002$, mean error rates were 9.03% and 50.58% for the with-example and without-example conditions, respectively), which does not lend support to such account.

## Discussion

The current result showed that the provision of an exhaustive example of the way in which a director's perspective constrains reference led to dramatically lower rates of egocentric errors. This suggests that the revised instructions likely helped participants to use the director's perspective effectively. We do not think that the with-example condition merely presents an improved instruction, as participants in this condition were not only quicker to produce a first correct response, they were also consistent at maintaining a lower (but above floor) level of egocentric errors since the first correct response. Furthermore, performance following "standard" instructions (without the example of the full chain of inference) was previously found to correspond to social functioning profiles as measured by autistic and psychotic characteristics [22]. Individuals who score highly on either an autistic or psychotic characteristic were less

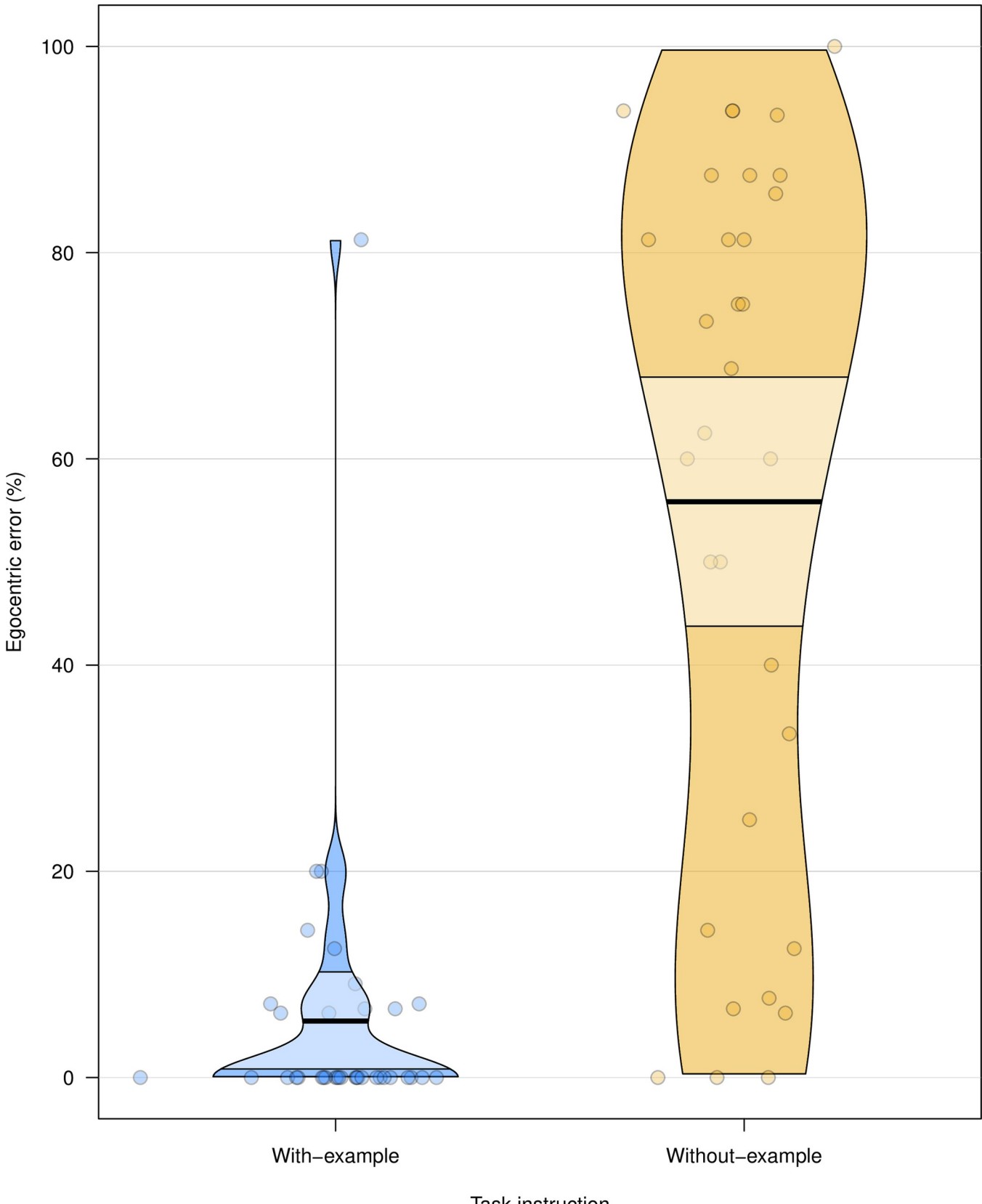

**Fig 2. Pirate plot for the percentage egocentric errors from Experiment 1.** Each circle represents the mean percentage egocentric error for a participant. The bold horizontal lines correspond to the condition means, the light-coloured bands around the means correspond to the confidence intervals.

likely to succeed in considering a director's perspective compared to individuals who score evenly on the two characteristics. This suggests that social functioning may account for communicators' propensities to infer the relevance of their communicative partners' perspectives in the absence of any additional incentives.

It is worth noting that the with-example condition served two functions. Firstly, it highlighted the way in which participants' own perspective led to an incorrect response "... you might be tempted to move this object, but this object isn't actually available to her...". Therefore participants may have been prompted to inhibit the use of their own perspective to interpret the director's utterance. Secondly, it highlighted the way in which the director's perspective should be used to interpret her utterances "... she must be talking about this object instead, because this is an object that she can see and can talk about...". To this end, participants may have been prompted to infer the ways in which the director's perspective should be used to interpret her utterance. It is noteworthy that successful implementation of either step would lead to better performance. This is because a full inhibition of self-perspective would lead participants to only consider objects in the common ground as potential referents, resulting in no egocentric errors. On the other hand, a correct inference about the ways in which the director's perspective should be used would also lead to correct interpretation of her utterance. Therefore the current experimental finding could be explained by either successful self-perspective inhibition or other-perspective-use, or both. In other words, the current experiment's exhaustive example may not be necessary. It is possible that just one of the components of the example is critical for improving the use of the Director's perspective. Experiment 2 was designed to disentangle the effects of self-perspective inhibition and other-perspective-use. It also served as a replication of the striking reduction in egocentric errors observed in Experiment 1 without the variation in the magnitude of common ground.

## Experiment 2

In the current experiment, four versions of overt instructions were employed, each placing selective emphasis on self-perspective inhibition versus other-perspective-use. The four versions of instructions were constructed in a factorial manner so that the effects of self-perspective inhibition and other-perspective-use can be fully dissociated. It is possible the two effects to be individually effective in boosting ToM-use. It is also possible that a combined effect is necessary to increase the propensity of ToM-use, in which case an interaction effect will be observed.

**Participants.**   One hundred participants (19 males, mean age 20.41 years, age range 18 to 29 years) gave informed consent to participate in the study and were tested by three experimenters at the Lancaster University. We expect the effect sizes for the current experiment to be smaller than Experiment 1, as we are attempting to separate the contributions of the two instruction steps. G*Power indicated that 76 participants were needed to detect an interaction effect, with power set to 0.8 and assuming an effect size of 0.2. Therefore 100 participants were recruited to ensure sufficient power is achieved. Six participants were replaced as they self-reported strategies unrelated to ToM-use.

**Design & procedure.**   A 2 x 2 x 2 mixed design was employed with condition (control, experimental) as a within-participant variable, other-perspective-use (other-explicit, other-not-explicit) and self-perspective-inhibition (self-explicit, self-not-explicit) as between-participant variables (see Table 1 for wording of instructions). Each participant was assigned to one

of four versions of overt instruction. The remaining aspects of the design was identical to Experiment 1, with the exception of the number of objects on the grid in the current experiment was fixed at 8, so that it matched the complexity of the grid employed in a widely employed version of the director task [2].

## Results

Trials with response timeout were excluded prior to analysis, leading to exclusion of 7.13% of the critical trials from the current experiment. The analysis strategy was identical to that of Experiment 1, with the following exceptions. The fixed effects were other-perspective-use (other-explicit, other-not-explicit) and self-perspective-inhibition (self-explicit, self-not-explicit). Participant, grid image, reward (cash, course credit), and experimenter (RA 1, RA 2, RA 3) were included as random effects. Models with only participant and grid image as random effects were highly similar to the models that had included reward and experimenter as random effects. The fitted model contained intercepts for all random effects, and random slopes for self-perspective-inhibition by grid image, the interaction between self-perspective-inhibition and other-perspective-use by grid image.

The two forms of reward were not designed to provide differentiating incentives for performance. Both forms rewards were advertised as compensation for participants' time rather than direct incentive for performance. Nonetheless, to check whether this random effect significantly alters the model fit, we compared a model with versus without the random effect of reward. Comparison showed no significant difference between the two models, $\chi^2 = 0.03$, df = 1, p = .864, $BF_{01} = 37.04$. Bayesian factor ($BF_{01}$) was calculated to quantify evidence for a null model relative to an alternative model. The null model includes all fixed and random effects apart from the random effect of reward. The alternative model which additionally includes a random effect of reward was compared against the null model. The Bayes factor was calculated from the Bayes information criteria (BIC) obtained from the null and alternative models [31]. The BF01 indicates that the alternative model was 37.04 times less favourable than the null model. This suggests that reward is very unlikely to alter the model fit.

The number of trials to first correct response and error rate following first correct response were aggregated by grid image and participant, therefore it was not possible to include either terms in the mixed models. These models had reward and experimenter as random effects. Maximal models were fitted to both the number of trials to first correct response and error rate following first correct response. Both models contained intercepts for reward and experimenter, slopes for self-perspective-inhibition by reward, other-perspective-use by reward, the interaction between self-perspective-inhibition and other-perspective-use by reward, self-perspective-inhibition by experimenter, other-perspective-use by experimenter, and the interaction between self-perspective-inhibition and other-perspective-use by experimenter (for a summary of all analyses for the current experiment, see Table 4).

Effects of other-perspective-use on percentage egocentric errors and number of trials to first correct response were observed (other-explicit < other-not-explicit, see Fig 3). The instruction to inhibit participants' own perspective did not have a significant effect on egocentric error ($BF_{01} = 27.78$). The interaction between self-perspective-inhibition and other-perspective-use was not significant ($BF_{01} = 15.63$). The linear mixed effects models analysis on response times only showed a significant effect of condition, $\chi^2 = 4.04$, df = 1, $p = .044$ (control < experimental).

**Replication of Experiment 1.** Direct comparisons between the other-explicit-self-explicit condition (equivalent to the with-example condition in Experiment 1) and other-not-explicit-self-not-explicit condition (equivalent to the without-example condition in Experiment 1) showed clear replication across all dependent variables: percentage egocentric error ($p < .001$,

**Table 4. Summary of mixed models from E2.**

|  | β | SE | χ2 | df | p |
|---|---|---|---|---|---|
| **Egocentric error** | | | | | |
| self | 0.61 | 0.77 | 0.63 | 1 | 0.428 |
| other | 2.12 | 0.76 | 7.69 | 1 | **0.006** |
| self*other | 1.80 | 1.55 | 1.78 | 1 | 0.182 |
|  | β | SE | χ2 | df | p |
| **Number of trials to first correct response** | | | | | |
| self | 0.99 | 0.53 | 2.96 | 1 | 0.085 |
| other | 1.66 | 0.52 | 4.12 | 1 | **0.042** |
| self*other | 0.56 | 1.55 | 0.13 | 1 | 0.721 |
|  | β | SE | χ2 | df | p |
| **Error rate following first correct response** | | | | | |
| self | -0.04 | 0.05 | 0.45 | 1 | 0.500 |
| other | -0.13 | 0.08 | 1.48 | 1 | 0.225 |
| self*other | -0.13 | 0.17 | 0.56 | 1 | 0.453 |

other-explicit-self-explicit < other-not-explicit-self-not-explicit), number of trials to first correct response ($p = .001$, other-explicit-self-explicit < other-not-explicit-self-not-explicit), and error rate following first correct response ($p = .014$, other-explicit-self-explicit < other-not-explicit-self-not-explicit). Additionally, the error rates following first correct response across all conditions were significantly higher than floor level ($t$s > 3.05, $p$s < .006; mean error rates ranging from 14.96% to 35.71%), once again providing no evidence for the alternative account that improved performance was merely driven by clearer instructions.

## Discussion

The current experiment played a critical role in disentangling the respective effects of self-perspective inhibition and other-perspective-use. Results showed that selective emphasis on the way the director's perspective constrained reference clearly helped participants to infer the correct ways in which to use the director's perspective. This suggests that the ways in which the director's perspective constrains reference is the processing step participants most likely neglected, and hence benefitted from being prompted via explicit instructions.

Interestingly, the instruction to inhibit the use of one's own perspective had no significant effect on measures of egocentrism. One possible explanation is that one's own perspective is salient and frequently used, therefore it cannot be easily modulated via instructions. However, it is unlikely that participants lack the requisite cognitive ability to inhibit their own perspective. As shown by Apperly et al. [2], participants performed with much greater accuracy on a director task when instructed to adopt a rule-based strategy to "discount all slots with grey background" compared when instructed to "take the director's perspective into account". Curiously, the current results suggest that even when specifically instructed to inhibit their own perspective, participants do not appear to spontaneously adopt such a simple discounting strategy. In contrast, a dramatic improvement in perspective-taking performance was seen when an exhaustive instruction to both inhibit self-perspective and to use the director's perspective was given. This highlights the possibility that self-perspective inhibition may need to be understood in the full context of perspective-taking. It is not sufficient for participants to be instructed to not use their own perspective if they do not know whose perspective they ought to adopt and how to do so, which may boil down to being sufficiently incentivised to invest the cognitive effort. Relatedly, the current results suggest that instructions for participants not to

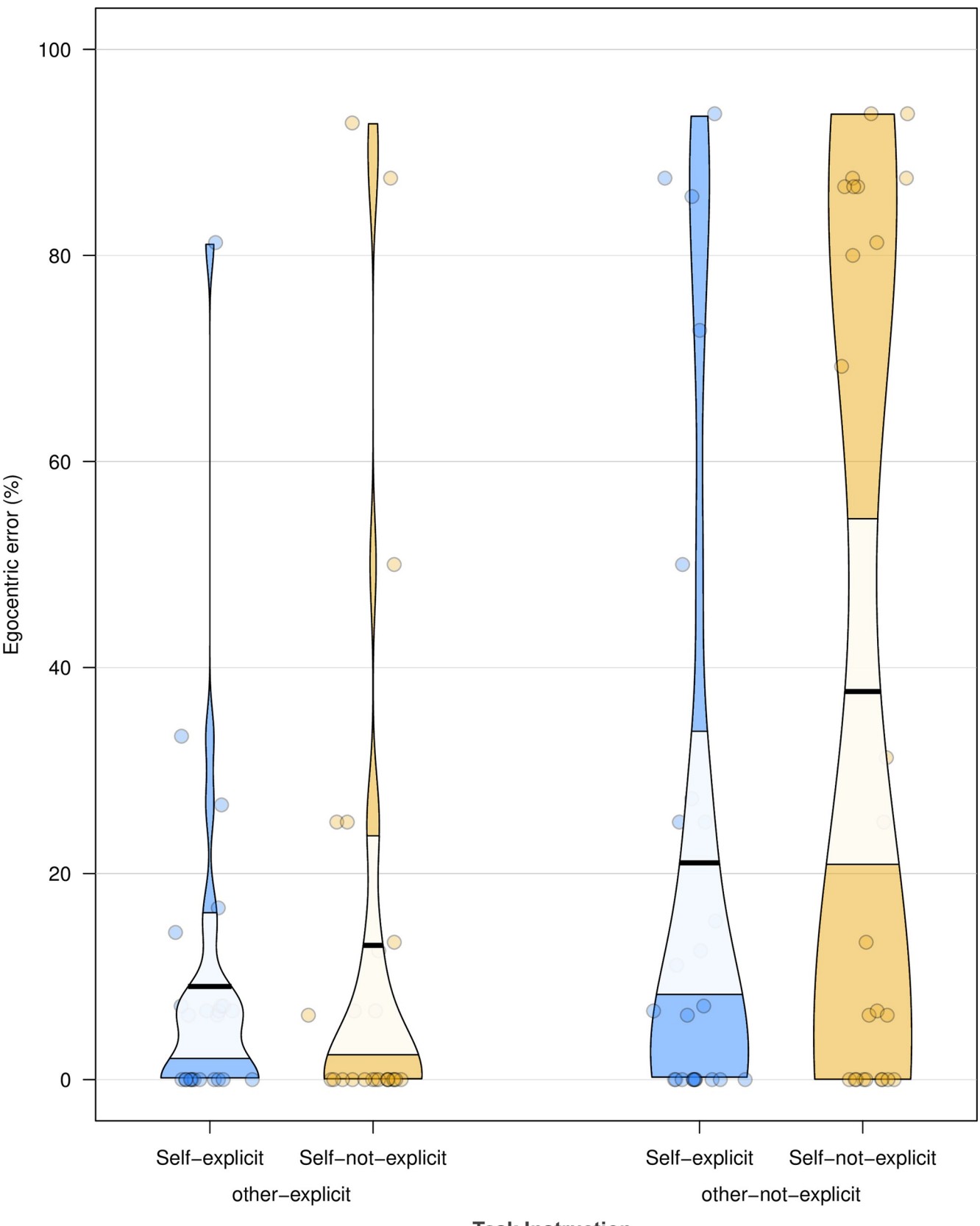

**Fig 3. Pirate plot for the percentage egocentric errors from Experiment 2.** Each circle represents the mean percentage egocentric error for a participant. The bold horizontal lines correspond to the condition means, the light-coloured bands around the means correspond to the confidence intervals.

use their own perspective to interpret the director's utterance did not sufficiently incentivise them to take the next inferential step and work out a way to use the director's perspective. This further confirmed that inference about the ways in which the director's perspective constrains reference is the processing step most likely neglected, as it was still overlooked when a different part of the inference steps was modelled in the instructions.

## General discussion

The current findings demonstrated that listeners tend to overlook the inferential steps required to successfully use speakers' perspectives, but can do so when prompted via instructions. Experiment 1 showed that the provision of a clear introductory example of the full chain of inferences required to successfully use a speaker's perspective led to large improvement in listeners' performance. Experiment 2 provided a replication of the key contrasts in Experiment 1, and further specified that the inferences about using a speaker's perspective, but not inhibiting the listeners' own perspective, played a significant role in lowering rates of egocentric errors. These results suggest that listeners do not need to be provided with an example of the full chain of inference in order to perform at a high level. Instead, listeners benefitted specifically from a prompt to identify precisely how a speaker's perspective constrains reference.

### Modelling inferential steps required for successful ToM-use

The current findings replicated previous observations that when listeners were simply instructed to take a speaker's perspective into account without specific instructions to inhibit their own perspective or use the speaker's perspective, they commit high rates of egocentric errors. This indicates that in the context of this experimental task, the propensity for listeners to invest the cognitive effort to take these inferential steps is low. In contrast, when small but systematic manipulations of the overt instructions were employed to model the inferential steps required to use the speaker's perspective, performance improved considerably. This suggests that the provision of an example of the inferences required to use the speaker's perspective was sufficient to prompt successful ToM-use.

Such effects were not only observed in rates of egocentric errors, they were also seen in the number of trials to first correct response across both experiments. This indicates that the revised instructions not only reduced rates of egocentric errors overall, it also made participants faster to implement the correct strategy of taking account of the director's perspective. Taken alone, this finding might suggest that participants simply found it easier to understand that they should take account of the director's perspective when given the revised instructions. Importantly, we think this may not be the full story, because the revised instructions also reduced participants' error rate after their first correct response (in Experiment 1), and these egocentric errors after a first correct response were not eliminated (in either experiment). This is not the pattern that would be expected if using the director's perspective was perfectly easy for participants once the revised instructions had enabled them to perform the task in the way intended. Instead we suggest that the revised instructions may have incentivised participants, via concrete examples, to make the required inference about using the speaker's perspective until they have successfully resolved a critical reference.

Interestingly, effects of instruction on other-perspective-use were seen in the overall rates of egocentric error and the number of trials to first correct response, but not in error rate

following first correct response. Notably, this measure did capture differences between the "all or nothing" conditions (the with-example versus without-example conditions in Experiment 1 and the other-explicit-self-explicit versus the other-not-explicit-self-not-explicit conditions in Experiment 2). This suggests that highlighting the ways in which the speaker's perspective constrains interpretation of their instructions may not on its own offer sufficient incentives for participants to use the speaker's perspective consistently, making it possible to observe an additional benefit from being prompted to ignore one's own perspective.

The current study identified two possible sub-processes in the *use* stage of the Apperly [26] mindreading model. In order for listeners to successfully *use* what they know about a speaker's perspectives, they need to inhibit information privileged to themselves, and infer how the speaker's perspective constrains interpretation of their message. The current finding that prompts for listeners to use a speaker's perspective significantly reduce rates of egocentric errors indicates that listeners were unlikely to make such inferences spontaneously. In contrast, prompts for listeners to inhibit their own perspective did not significantly lower rates of egocentric errors overall. We assume that self-perspective-inhibition is nonetheless integral to ToM-use, but that these results indicates that prompting participants to inhibit their own perspective had little overall effect, as discussed in Experiment 2. The dissociated effects of self-perspective inhibition and other-perspective use could help explain the high rates of egocentric errors observed in previous studies [2,3], and potential reasons for the great difficulty in lowering rates of egocentric errors. It seems likely that listeners know that they need to account for the speaker's perspective in some way, and they know that this implies that they should not use their own perspectives to interpret the speaker's utterance. However, not all listeners spontaneously inferred *the ways in which the speaker's perspective constrains interpretation of her instruction.*

## Variability in ToM-use

The current findings revealed that consistency in ToM-use cannot be taken for granted. This echoes recent work which suggests that the context and saliency of the cues associated with self and other could alter the ways in which perspectives are inferred and considered over the course of an interaction [32–34]. Additionally, a conversational partner and their social relationship with the listener could determine the degrees to which listeners are motivated to take their partner's perspective into account [35]. Furthermore, specific conversational goals could also affect the degrees to which speakers display sensitivity towards their listener's perspective [36]. Unlike conceptual tests of ToM, which typically delivers a pass versus fail verdict, ToM-use clearly varies according to a wide range of contextual factors. On the one hand, this may fit with our intuition about the variability of everyday social interactions. Faux pas do occur, yet they do not occur all of the time nor do they always occur to the same individual. On the other hand, the variability in ToM-use makes for a difficult theoretical debate. It is possible that the ongoing debate about whether listeners readily take a communicative partner's perspective into account is unnecessarily dichotomous. It may be more productive to identify the context in which communicators succeed or fail to account for others' perspective. Additionally, with large degrees of variability in the experimental paradigms employed and the task instructions delivered, it is critical for future work to systematically report full instructional materials, so we can begin to understand the ways in which communicators may be incentivised to take others' perspectives.

## Conclusion

The current findings clearly demonstrated that listeners do not readily make inferences about using speakers' perspectives, but can do so when prompted. Successful referential

communication requires listeners to infer that 1. they should not use their own perspective to resolve reference 2. the specific ways in which a speaker's perspective constrains reference. Two experiments showed that an example of the full set of inferential steps required led participants to much greater levels of success in referential communication. Furthermore, specific prompt to use the speaker's perspective on its own was effective in boosting performance. The current findings suggest that inference about the ways in which others' perspectives need to be used is likely to be the primary obstacle to successful referential communication. Furthermore, simple manipulations such as an introductory example of the inferential steps required, can incentivise listeners to invest the cognitive effort to overcome such obstacles. Finally, the current findings provide an important foundation for advancing our knowledge about individual differences in mindreading in real life. A fruitful future direction would be to investigate the interactions between individual differences in the propensity to make the required inferences at the right time, cognitive flexibility (e.g., [10], and social functioning (e.g., [22,35]). Such insights will bring us closer to understanding the successes and failures in everyday mindreading.

## Supporting information

**S1 File. RT analysis.**
(DOCX)

**S1 Appendix. Complete list of critical instructions.**
(DOCX)

## Acknowledgments

The authors would like to thank Beth Armstrong for her help with data collection.

## Author Contributions

**Conceptualization:** J. Jessica Wang, Ian A. Apperly.

**Data curation:** J. Jessica Wang.

**Formal analysis:** J. Jessica Wang.

**Funding acquisition:** J. Jessica Wang, Ian A. Apperly.

**Investigation:** J. Jessica Wang, Natalia Ciranova, Bethany Woods.

**Methodology:** J. Jessica Wang, Ian A. Apperly.

**Project administration:** J. Jessica Wang, Natalia Ciranova, Bethany Woods.

**Resources:** J. Jessica Wang.

**Software:** J. Jessica Wang.

**Supervision:** J. Jessica Wang.

**Validation:** J. Jessica Wang.

**Visualization:** J. Jessica Wang.

**Writing – original draft:** J. Jessica Wang.

**Writing – review & editing:** J. Jessica Wang, Ian A. Apperly.

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
