## [Decision Letter · Decision Letter 0]

9 Mar 2020

PONE-D-20-01714

Why are listeners sometimes (but not always) egocentric? Making inferences about using others’ perspective in referential communication

PLOS ONE

Dear Dr Wang,

Thank you for submitting your manuscript to PLOS ONE. After careful consideration, we feel that it has merit but does not fully meet PLOS ONE’s publication criteria as it currently stands. Therefore, we invite you to submit a revised version of the manuscript that addresses the points raised during the review process.

I carefully read the manuscript myself and asked two reviewers, both experts in perspective-taking and referential communication to evaluate the quality of the manuscript. The reviewers provided what I consider to be excellent and fair critiques, as well as several suggestions for improvement. Based on the reviewers’ comments, their ratings, and my own evaluation of the manuscript, I am pleased to offer you an opportunity to revise and resubmit an updated manuscript for potential publication in PLOS One. Note that this invitation to revise does not constitute a guarantee of eventual acceptance. If you choose to revise the manuscript, I will likely send it to the same reviewers to reevaluate. I also ask that you please provide point-by-point responses to each comment, either describing changes made in response to the comment or explaining why no change was made.

For the content of the reviews, I will not repeat all the various issues as they speak for themselves, but I will focus on a few issues that were particularly salient. 

Both reviewers question the theoretical claims that are being made. Reviewer 1 would like to know more about the motivation and background that drove this work and Reviewer 2 wonders what is exactly meant by claims concerning the heart of the motivation, this idea of “full chain of inferences.” In my own reading, I too wanted a clearer accounting of what you exactly meant by the inferential steps that are being examined and a more explicit foreshadowing of how these steps are being manipulated within your paradigm/instructions. It seems that what you mean by “full inferential” (or “full chain of inferences”) is this idea of “other-explicit” and “self-explicit,” that is, a speaker has a unique perspective that needs to be taken into account and the self has a unique perspective that might been to be overridden. In the Introduction, the only inferential step that is emphasized is the “other-explicit.” It is only when the reader gets to the Methods that what is meant by full inferential steps (at least I think) is made clear - and really, it was only when reading through Table 1 and 2 captions that the ideas began to fall into place. 

Reviewer 2 also wonders whether their understanding of the results of Experiment 2 - as being a sequential step of inference - is correct, and if so, greater discussion of what this means in terms of cognitive processing is needed.  I share with Reviewer 2 a concern about whether my understanding of the claims is correct and what this means for interpretation. For example, there is a claim repeated in the Discussion that having the full chain of inferences required to use the speaker’s perspective was sufficient to prompt successful ToM-use. But based on the results from Exp 2, I do not quite see how this can be true. Do not the results show that having just an other-explicit component produced a similar pattern of results as having both other-explicit and self-explicit components in the instructions? If so, then having examples with a full chain of inferences is not required (as stated on line 447 and in the Conclusion section). It is useful but not absolutely necessary.  

I should note that when reading Reviewer 2’s point above, it also made me think of how this all relates to the theoretical framing used in the Introduction, namely the section on p. 7 lines 130-145 on Apperly’s three constituent stages. I would like to see how this earlier theoretical framing helps informs the current results. In the Introduction it is stated that previous studies observed egocentric errors at the “use” stage of Apperly’s model, but it seems that something else is being argued here for the current paper. The issue instead is about “calculating” not just others’ mental states (other-explicit) and explicitly recognizing one’s own perspective as being a potential distractor (self-explicit). But I am unclear on these finer theoretical points and how it relates to a full chain of inference. 

There were also potential study design concerns that raised red flags for interpretation. For example, Reviewer 1 notes that in the error rate is very high in the instruction condition without examples. Reviewer 1 asks whether the error rate is based on participants' final selection. This question dovetails with Reviewer 2’s astute observation that it is unknown how correct object selection (and what constituted an error) was operationalized. Another red flag brought up by Reviewer 1 is why both object size and color were varied within targets and distractors. I agree that this makes the correct (ToM) selection harder to process and could be contributing to the high error rate. And lastly, Reviewer 2 asks for greater clarification on whether the control and experimental versions were in the same block as this definitely could confound results. Where limitations exist, please be sure to make corrections and/or address them thoroughly in the Discussion. 

Reviewer 2 also makes the excellent point about this issue of the sensitivity of your measures to fully capture the cognitive dynamics that seem to be of interest in this paper. That is, you are interested in how response behavior changes over time. I strongly encourage you to consider adding trial ID as a fixed effect as I agree it would give you much better insight (perhaps error rates drop more dramatically over time in the full inferential sequence instructions than in the partial inferential sequences). But whether you add or not is not a condition for acceptance, but it would be good to have some speculation on this matter in the Discussion. 

In conclusion, although there are several major issues that need to be addressed, I hope they can be resolved because the topic is very interesting and I think this work can make an important contribution to the field. 

Again, please address every concern raised by the reviewers, as well as my specific comments that follow below. I look forward to seeing your revised version. Thank you. -Nick Duran

MY OWN ADDITIONAL COMMENTS:

Plos One puts great value on data availability and transparency in the code for statistical analysis. Perhaps I missed it, but is this information made available to the reader who might be interested in replication or even checking the statistical models for accuracy? Such data could be made available on an OSF repository (osf.io).

**Exp 1.**

For clarification, it would be good to integrate into the main prose the exact nature of the manipulations mentioned in the Table 1 and 2 captions. 

I would be interested in seeing the specifications within G*Power for determining sample size for Exp 1 and Exp 2. Given statistical models that involve between-subjects factor and interaction effects, both studies seem underpowered. However, I am more than willing to concede that I am making incorrect assumptions about how you determined sample size. 

What was the original rationale for the control condition in this study? It was eventually dropped from analysis because participants made so few errors, but curious why errors were even expected to begin with? 

p. 12, line 235: Please provide further explanation for why a large number of trials with correct responses were lost to high rates of error. What is the nature of this error?

p. 13, line 245: what exactly is the contrast coding scheme used?

p. 14, line252: what statistical test did you use to do this comparison?

In Exp 1 Discussion, I could not follow the argument being made about social functioning profiles (around line 287). Is this to make an argument that mere instructions are sufficient for bringing greater awareness/relevance to a director’s perspective? Please clarify in the prose.

**Exp 2.**

For setting the effect size, you state that you expect a smaller effect size from Exp 1. Why then choose an effect size of 0.6 for Exp 2 when it was set at 0.2 for Exp 1?

What was the rationale for fixing the grid size to 8 in Exp 2 when it was allowed to vary across a range for Exp 1?

The alternative hypothesis on page 20, starting line 383, is hard to follow and it took me some time to work out. I think what is being claimed is that an ego-centric bias was still compete with responses even with the clearest of instructions. Is this right? And this appears to be evidenced with the more detailed instruction (other- and self-explicit) where the error rate following first correct response is still high (above 0) and does not vary from that much from the less detailed instructions? It would be useful to know which numbers are being compared from Table 2 to make these claims. Also, the introduction of the alternative hypothesis arrives out of the blue. I would recommend introducing the alternative hypothesis earlier as a target for analysis/interpretation (rather than doing it in the Results section), or treat it as a standalone issue/explanation as you do with the section “Replication of Experiment 1.”  

We would appreciate receiving your revised manuscript by Apr 23 2020 11:59PM. To enhance the reproducibility of your results, we recommend that if applicable you deposit your laboratory protocols in protocols.io, where a protocol can be assigned its own identifier (DOI) such that it can be cited independently in the future. For instructions see: http://journals.plos.org/plosone/s/submission-guidelines#loc-laboratory-protocols

We look forward to receiving your revised manuscript.

Kind regards,

Nicholas D. Duran

Academic Editor

PLOS ONE

Journal Requirements:

2. We note that Figure 1 includes an image of a participant in the study. 

Reviewers' comments:

Reviewer's Responses to Questions

**Comments to the Author**

1. Is the manuscript technically sound, and do the data support the conclusions?

Reviewer #1: Partly

Reviewer #2: Yes

2. Has the statistical analysis been performed appropriately and rigorously? 

Reviewer #1: Yes

Reviewer #2: Yes

3. Have the authors made all data underlying the findings in their manuscript fully available?

Reviewer #1: Yes

Reviewer #2: Yes

4. Is the manuscript presented in an intelligible fashion and written in standard English?

Reviewer #1: Yes

Reviewer #2: Yes

5. Review Comments to the Author

Reviewer #1: This research examines the role of explicit instructions on listeners’ perspective-taking in the referential communication task. When the director’s scene and the listener’s scene differ, the listener needs to consider the common ground rather than taking their privileged ground. Previous studies have shown that listeners are not always successful to take the director’s perspective to identify a target following the director’s instruction. In two experiments, the authors manipulated the types of instructions and tested listeners' perspective-taking. The results showed that listeners made fewer egocentric errors when they were given explicit instructions with examples (E1). Especially, explicit instructions on using the director’s perspective modulated listeners’ performance, but instructions on inhibiting their privileged perspective did not help (E2). Thus, the findings showed that listeners can take the director’s perspective when they are prompted with appropriate instructions.

I have a few comments and questions below.

First of all, I was not fully convinced with the motivation of this research. It is not quite surprising to see participants’ performance improved with explicit instructions. In introduction, it needs to build theoretical motivation and background that drove this research.

Especially, providing explicit instructions in this research may lack ecological validity. In everyday life, we are not told about perspective differences between partners. We naturally notice it and integrate that information in language processing. I think it is OK to use manipulations that lack ecological validity, but it needs justification. In that sense, it would be helpful to read about theoretical motivations about this research question/manipulations in introduction (and theoretical implications in discussion as well).

Secondly, in this sort of experimental paradigm in which perspective-taking is tested, researchers have manipulated only the “size” of critical items, the color remained the same (Ryskin et al., 2015). However, in this research, for example in Fig 1, the colors of the three presents are different and I think this is problematic. People use color adjectives all the time, and especially in Fig 1, speakers would not describe the target item using a size adjective. They would be highly likely to describe it using a color adjective (e.g., the purple present). If so, when participants heard the size adjective, it is just difficult for them to process. Why would you use a difficult size adjective when you can simply use a color adjective? I was wondering if all items were like in Fig 1 (critical items with different colors)? Or do critical items in Fig 1 and Table 1 happen to have different colors? It needs to be clarified.

Thirdly, the results are quite straightforward, but I was surprised by a high rate of errors that participants made especially in the condition without examples (E1 and the comparable condition in E2). Was it based on their final selection? Then, this means that more than half of the time, participants made egocentric errors in E1 (without-example). I understand that listeners consider egocentric knowledge in the middle of processing but it’s surprising that they made a final selection more than half of the time based on their egocentric knowledge. It would be great to clarify the dependent measure and I also wonder if there was a procedure to confirm that participants understood the experimental setting (e.g., what the green background indicated…).

Regarding the dependent measure, in the results, the authors analyzed the response time (pg. 14) as well as other measures (egocentric error, # of trials to first correct response, and error rate following first correct response), but descriptive stat of response time was not reported.

Minor points:

In E2, the authors included grid image, reward, and experimenter in the model. To make this consistent across experiments, it needs to be done in E1 as well. (Or if it is already done, it needs to be reported.)

In Table 1, the condition labels were not clear. I had to read back and forth to understand each condition. If it can be clarified, it would be helpful.

Reviewer #2: Summary: Across two experiments, the authors examine the effect of instructions on the distribution of egocentric responses in the "director task". The findings suggest that instructions about the inferential steps of assessing the director's perspective reduce the egocentric errors. Experiment 2 clarifies that it is the instructions' emphasis on the use of the task partner's perspective that drives this effect.

I have reviewed this article before at another journal. In this review, I focus my assessment on the extent to which the authors addressed my previous comments adequately in this submission. As I noted in my previous review, overall, I appreciate the novelty of the study in disentangling the effect of instructions to inhibit one's perspective vs. to use the task partner's perspective.

The authors effectively addressed one of my earlier comments about the theoretical framing (namely, about instructions previously being framed as motivating listeners to make inferences about the speaker’s perspective). Still, a number of my previous comments persist, including methodological and analytical points that could have been easily clarified.

I present these pending methodological issues next:

1. Please be explicit about whether participants saw both the control and experimental version of an otherwise identical display (i.e., whether they saw both displays shown in Figure 1). If so, I assume that critical and experimental versions were not in the same block, since this contrast could highlight the disparity between self and other perspectives and could confound the results. In either case, please specify how items were distributed across trials and blocks.

2. The distinction between conditions (control, experimental) is opaque upon first mention (p. 9), and is not fully described until p. 11. Please establish early on that these terms have to do with the absence vs. presence of a distractor in the trial.

3. It’s not clear initially, on p. 9., what the "magnitude of common ground" means (“3, 5, 7, 9”). On p. 13 the authors suggest that these values refer to "slots", but whether they are slots in common ground or NOT in common ground remains unclear. (Figure 1 suggests that it's the latter, seeing that there are 5 occluded and 11 open slots.) Please clarify.

4. It's not clear what would constitute egocentric errors in the control condition (p. 12, line 222). If the correct selection is the same object from both the participant's and the director's perspective, I don't follow how there could be an egocentric error to even yield the "2 egocentric errors” observed in the control condition across the two studies. Did the authors count the selection of the irrelevant object as an egocentric error (under the rationale that it is at the same location as the distractor/egocentric item of a corresponding display)? That doesn’t seem to be a felicitous interpretation of “egocentric”.

5. Please specify how correct object selection and response times were operationalized. Was response time based on the first click on the computer screen? on the release of the drag-and-drop gesture? on the final object selection or drag-and-drop release? All of these definitions are plausible. What counted as correct object selection: clicking anywhere in the cell of that object or at a more circumscribed area? Please clarify how data was collected from participants' interaction with the displays.

Some persisting theoretical points:

1. The authors refer to the instructions as describing the "full chain of inferences". What is that chain of inferences exactly? Is the chain of inferences: “Can she see my largest present? If yes, select. If not, which is the next largest present?” Or something else? In the Conclusion section, the authors present two inferences: “1. [Listeners] should not use their own perspective to resolve reference, 2. the specific ways in which a speaker’s perspective constrains reference.” Is this the chain of inference? If so, framing this as a chain suggests a sequential order. And if that’s the claim, the authors need to be explicit about how their results from Experiment 2 support it.

2. In this submission it’s not as clear what the rationale is of choosing the 3 DVs (overall rate of egocentric errors, number of trials to first correct response, and error rate following first correct response). The inclusion of the two latter metrics suggests that the first correct response reveals something important about the listener’s cognitive dynamics throughout the experiment. But the authors don’t explain how and why. Much later in the paper (p. 20, line 383-386) the authors suggest that they use these measures to test the alternative hypothesis that the instructions fully clarified what participants should do. The connections between the choice of these measures, the authors’ preferred hypothesis, and this alternative hypothesis need to be clarified.

As I recommended previously, instead of or in addition to these coarser-grained DVs, the authors could include trial id as a fixed effect in their models. Such an exploratory analysis could offer insights about the stabilization of other-centric responding over time under the different instruction conditions.

A related point: the authors suggest that, because the listeners’ error rates were substantially different from 0% after their first correct response, listeners “did not simply grasp what they were supposed to do” after the instructions. I’m not sure what “grasping what they were supposed to do” means here. But as I pointed out before, perspective-taking is not all-or-none; it need not be invariably at floor or at ceiling.

I leave it up to the Action Editor whether the authors should conduct additional analyses to examine at the distribution of egocentric errors over time in finer grain. But given the authors' theoretical points about sub-stages in perspective-taking and chains of inference in resolving reference, I suggest (as I did before) that they consider related work on how perspective-taking choices evolve and stabilize over time, not just within trials but across trials, over the course of an interaction (e.g., see Duran & Dale, 2014; Duran & Dale, 2011; Dale, Galati, et al., 2019; Galati, Dale, and Duran, 2019).

Small points

• p.4: Thanks for unpacking and future-proofing the Hillary Clinton example. Just a typo: “winner” -> “winning”

• p. 12, line 222: “excluded prior analysis” -> “excluded prior to analysis”

• Please replace R and lmer syntax with words in the text (namely, avoid | and * ).

• Footnote 3: incentivises -> incentives

• The two Experiments don't report results consistently. E2 includes Bayes factors, but E1 does not. Please make them consistent.

Alexia Galati

6. PLOS authors have the option to publish the peer review history of their article (what does this mean?). If published, this will include your full peer review and any attached files.

Reviewer #1: Yes: Si On Yoon

Reviewer #2: No

---

## [Author Response · Author response to Decision Letter 0]

9 Jul 2020

Please see attached 'response to reviewer' document

---

## [Decision Letter · Decision Letter 1]

7 Sep 2020

PONE-D-20-01714R1

Why are listeners sometimes (but not always) egocentric? Making inferences about using others’ perspective in referential communication

PLOS ONE

Dear Dr. Wang,

Thank you for submitting your manuscript to PLOS ONE. After careful consideration, we feel that it has merit but does not fully meet PLOS ONE’s publication criteria as it currently stands. Therefore, we invite you to submit a revised version of the manuscript that addresses the points raised during the review process.

Thank you for this resubmission and the obvious efforts in addressing the comments and concerns from the first round of reviews. I believe the manuscript to be much improved because of these efforts, a sentiment shared by the reviewers. 

As you can see from the new reviews - which I thank the reviewers for doing with such care - there are still issues to be resolved. Many of these revolve around a theoretical framing that needs to be further clarified and strengthened. There is no requirement here to justify the importance of your theoretical stance, but merely to provide enough detail to remove ambiguity in how your conclusions relate to your data and a larger literature. Your manuscript is strong on all of PLOS ONE’s publishing criteria, except for one: "Conclusions are presented in an appropriate fashion and are supported by the data.” This is not a seriously damning weakness, but a few claims/conclusions need to be tempered given plausible alternative explanations and a referential communication task that has been questioned by others for its ability to genuinely capture perspective-taking behaviors in richer communicative scenarios. The suggestions for how to temper are provided by reviewers. I also urge you to seriously consider other suggestions for how to better draw out the novelty of your findings and their implications, as well as where greater precision in your interpretations could be made. 

As before, please address every concern raised by the reviewers by directly communicating what change was made to your manuscript because of the comment, or justification for why you did not choose to make a change.  I look forward to seeing your revised version. This paper should make a strong contribution once all points are addressed. Also know that I am committed to expediting this manuscript to the next stage assuming the necessary changes have been made.

Sincerely, Nick Duran

We look forward to receiving your revised manuscript.

Kind regards,

Nicholas D. Duran

Academic Editor

PLOS ONE

Reviewers' comments:

Reviewer's Responses to Questions

**Comments to the Author**

1. If the authors have adequately addressed your comments raised in a previous round of review and you feel that this manuscript is now acceptable for publication, you may indicate that here to bypass the “Comments to the Author” section, enter your conflict of interest statement in the “Confidential to Editor” section, and submit your "Accept" recommendation.

Reviewer #1: (No Response)

Reviewer #2: (No Response)

2. Is the manuscript technically sound, and do the data support the conclusions?

Reviewer #1: Yes

Reviewer #2: Yes

3. Has the statistical analysis been performed appropriately and rigorously? 

Reviewer #1: Yes

Reviewer #2: Yes

4. Have the authors made all data underlying the findings in their manuscript fully available?

Reviewer #1: Yes

Reviewer #2: Yes

5. Is the manuscript presented in an intelligible fashion and written in standard English?

Reviewer #1: Yes

Reviewer #2: Yes

6. Review Comments to the Author

Reviewer #1: This is a resubmission of work investigating the role of explicit instructions on ToM (taking the partner’s perspective) in a referential communication task.

My main concern in my previous review was related to the theoretical motivation/implication. The authors have edited the manuscript, and Experiments are now more clearly motivated. The manuscript is improved by the more detailed descriptions of the experimental manipulations and analyses in the paper.

I believe the paper is much clearer now although some issues could still be made and I have a few remarks on the revised manuscript.

- Now I understand the theoretical motivation better, but the paper downplays the literature that has shown that listeners readily take the partner’s perspective in language comprehension (Hanna et al., 2003; Brown-Schmidt, 2009, etc.). There have been theoretical debates between the two groups on listeners’ ability to use common ground (Barr, 2008; Keysar et al., 1998; 2003 vs. Hanna et al., 2003; Hanna & Brenna, 2007; Brown-Schmidt, 2009); While the authors described only one side of the literature, there is a great amount of work that demonstrated that listeners could quickly identify the target based on the speaker’s perspective (Hanna et al., 2003; Brown-Schmidt, 2009; Ryskin et al., 2015, etc). I think it’s fair to introduce both sides of the literature in the introduction.

While the current work seems to be more consistent with Barr and Keysar’s claim and at odds with the results of the work by the Tanenhaus group, Arperly’s theory and the current results may contribute to reconciling the two seemingly contradictory claims. I wonder if the authors have thought about this point and if so, include it in the discussion. I bet a wider range of audiences will be interested.

- One thing I’d like to comment is that in the current study, listeners benefitted from explicit instructions, but without explicit instructions with examples, they fail to take the partner’s perspective 40-60% of the time (in E1 and E2). Even with this failure, it may not mean that listeners are not able to take the partner’s perspective; they just don’t do without specific motivation or conversational goals. In Yoon, Koh, & Brown-Schcmidt, (2012), the rate of perspective-taking varied depending on the conversational goals (instruction vs. request). Like this, participants in the “without-example” condition might be able to do so as in the explicit instruction conditions, but they might not be motivated enough – which is one of the disadvantages of non-interactive referential communication tasks (compared to interactive version). The claim in the manuscript may need to be tempered.

- pg, 12, among 96 fillers, 24 contained scalar adjectives, and 14 contained non-scalar adjectives. I guess 58 fillers were bare noun phrases (E.g., bowl). This needs to be clarified. Along with this, I have a question regarding the fillers of scalar adjectives. In Figure 1 and Table 1, I do not see any size-contrast filler items. The only size-contrast item is the critical target item (e.g., torch, present). How were scalar adjectives used to refer to filler items? Listeners are sensitive to the use of adjectives; if speakers use an unnecessary adjective, their comprehension processing might be significantly affected (e.g., when there is only one bowl, and the speaker says “move the small bowl…”).

- In Table 2, please include descriptive statistics of the control conditions. I understand the authors only analyze the critical condition in the model, but it will be helpful to understand the entire data.

- in abstract and in the intro (pg. 4), the phrase “healthy adults have little trouble understanding others’ mental states” is confusing; does it mean they do not have trouble understanding that the partner can have “different” mental state or understanding the exact different view of the partner? For example, one can know that the partner can have a different view without knowing what exactly it can be vs. one can know exactly what the partner has in his view. It needs to be clarified.

- pg. 9, Experiment 2 further investigated whether both prompts are required, and whether they need to be given sequentially.  the later one was not tested.

- In Methods, were all participants native English speakers?

- pg. 26, line 564, “very few listeners spontaneously inferred…” This sentence is not consistent with data in E1 and E2. In Figures 2 and 3, many participants successfully take the partner’s perspective even in the “without-example” condition.

Reviewer #2: I have carefully read this revised manuscript, which I find considerably improved. You have addressed a number of my previous comments effectively, especially in terms of clarifying methodological points (e.g., how you operationalized correct responses and RTs, the rationale of your different DVs, how control and experimental trials were distributed, etc.).

I find that a few theoretical points can still be clarified further.

1. I appreciate your clarifications about the two inferential steps you identify as necessary for completing the director task successfully: (1) participants must ensure they are guided by the director’s perspective rather than their own (i.e., must inhibit their own perspective), 2. participants work out precisely how the director’s perspective constrains reference. However, I find that more work is needed to clarify these inferential processes further and strengthen the theoretical contribution of the paper.

First, I invite you to be more explicit about what step #2 involves in this director task. Specifically, I encourage you to be more specific about what you mean by “working out” how to use the director’s perspective. Is step #2 the recognition that the director cannot see items in a blocked shelf *AND* that, as a result, their intended referent of a scalar expression (e.g. “the short torch”) can be different from the participant’s?

Second, by extension, I invite you to be more explicit about how this “working out” of using the partner’s perspective can look like in other contexts involving perspective taking. Doing so can help clarify the connections between these findings and the more generalizable account you espouse, namely, Apperly’s model of mindreading (see point #2).

Third, in your response letter, you noted that you don’t consider these steps to be sequential, even though they happen to appear in sequence in the instructions. But this isn’t clearly discussed in the paper. In fact, the language used throughout the manuscript (“steps”, “inferential steps”, “chain of inference”) does suggest sequential steps. Please clarify. This is especially important because steps #1 and #2 together can result in a single heuristic for doing the task (e.g., “skip all blocked shelves”).

2. It’s not clear how the findings inform Apperly’s model of mindreading. In the Introduction you describe the two steps of the use stage of perspectival information (lines 158-165). In the Discussion you argue that your study identifies these two sub-processes of the use stage of mind reading (line 549). There’s circularity in this framing.

The most novel contribution of this work is the finding that highlighting step #1 with instructions (“inhibit your own perspective”) does not support perspective-taking performance, whereas highlighting step #2 with instructions does. This take-home message is lost a bit. This is in part because whenever the results of Experiment 1 are reported there is emphasis on the fact that that instructions with the “full chain of inferences” were required for successful perspective use. I suggest tempering these statements in your General Discussion, seeing that in Experiment 2 you actually find that an example with the full chain of inferences was not in fact required for successful performance.

Overall, I recommend highlighting your most notable contributions about the differential benefit of highlighting through instructions these two inferential steps and providing an explanatory account for this finding. That is: why is it that instructions to suppress the egocentric perspective don’t confer a performance benefit? Is it because adopting the egocentric perspective is fairly automatic and cannot be modulated by instructions? Addressing these points can bolster the theoretical contribution of the paper.

3. I don’t get the alternative account that you’re trying to rule out by testing egocentric errors past the first correct response against 0%. You argue that if participants fully understood the instructions through a more elaborated example, then floor-level error rates should be observed. I don’t see how understanding instructions obligates flawless performance. Human behavior is still replete with errors when there is response conflict, even when instructions are understood. What does “understanding instructions” mean? My general stance is that it is perfectly fine to conduct the test you did to establish that the obtained error rates are higher than zero. But I don’t see this alternative account as a plausible one that you need to rule out. If you retain this alternative account, I recommend unpacking why understanding instructions (which is different from applying the instructions) necessitates floor-level error rates.

Smaller points:

• You have changed your display in Figure 1 but, in the text, you still refer to “presents” rather than “torches” (p. 13).

• Please include in a footnote your explanation about including Bayes factors for null effects of interest.

• P. 25, bottom: You suggest that instructions “may have provided additional motivation for participants to make the required inference about using the speaker’s perspective…” This might be residual language from an earlier version of your paper. It’s not clear what you mean here about instructions serving as “motivation” or as a ‘sufficient prompt” (a few sentences below, p. 26 top).

Typos:

• P. 6, line 101: “not participants’ own perspective”  “not the participants’…”

• P. 6, line 106: “variations of the director task has”  have

• P. 8, line 166: “how people use mindreading information Apperly’s (2010).” ??

• P. 18, line 369: “who score highly on either autistic or psychotic characteristic”; add “an” or pluralize (“characteristics”)

• P. 20, line 422: “excluded prior analysis”  “prior to”

7. PLOS authors have the option to publish the peer review history of their article (what does this mean?). If published, this will include your full peer review and any attached files.

Reviewer #1: No

Reviewer #2: **Yes: **Alexia Galati

---

## [Author Response · Author response to Decision Letter 1]

23 Sep 2020

Please see attached file "Response to Reviewers"

---

## [Editor Report · Decision Letter 2]

29 Sep 2020

Why are listeners sometimes (but not always) egocentric? Making inferences about using others’ perspective in referential communication

PONE-D-20-01714R2

Dear Dr. Wang,

We’re pleased to inform you that your manuscript has been judged scientifically suitable for publication and will be formally accepted for publication once it meets all outstanding technical requirements.

Kind regards,

Nicholas D. Duran

Academic Editor

PLOS ONE

---

## [Editor Report · Acceptance letter]

14 Oct 2020

PONE-D-20-01714R2 

Why are listeners sometimes (but not always) egocentric? Making inferences about using others’ perspective in referential communication 

Dear Dr. Wang:

I'm pleased to inform you that your manuscript has been deemed suitable for publication in PLOS ONE. Congratulations! Your manuscript is now with our production department. 

Kind regards, 

on behalf of

Dr. Nicholas D. Duran 

Academic Editor

PLOS ONE